# Determining Perceived Self-Efficacy for Preventing Dengue Fever in Two Climatically Diverse Mexican States: A Cross-Sectional Study

**DOI:** 10.3390/bs12040094

**Published:** 2022-03-28

**Authors:** Esther Annan, Aracely Angulo-Molina, Wan Fairos Wan Yaacob, Nolan Kline, Uriel A. Lopez-Lemus, Ubydul Haque

**Affiliations:** 1Department of Biostatistics and Epidemiology, University of North Texas Health Science Center, Fort Worth, TX 76107, USA; mdubydul.haque@unthsc.edu; 2Department of Chemical and Biological Sciences/DIFUS, University of Sonora, Hermosillo 83000, Mexico; aracely.angulo@unison.mx; 3Faculty of Computer and Mathematical Sciences, Universiti Teknologi MARA Cawangan Kelantan, Kampus Kota Bharu, Lembah Sireh, Kota Bharu 40450, Kelantan, Malaysia; wnfairos@uitm.edu.my; 4Institute for Big Data Analytics and Artificial Intelligence (IBDAAI), Kompleks Al-Khawarizmi, Universiti Teknologi MARA, Shah Alam 40450, Selangor, Malaysia; 5Department of Health Behavior and Health Systems, School of Public Health, University of North Texas Health Science Center, 3500 Camp Bowie Blvd, Fort Worth, TX 76107, USA; nolan.kline@unthsc.edu; 6Department of Health Sciences, Center for Biodefense and Global Infectious Diseases, Colima 28078, Mexico; ulemus@cebgid.org

**Keywords:** knowledge, self-efficacy, dengue fever, prevention, mosquito control, Mexico, Sonora, Colima

## Abstract

Knowledge of dengue fever and perceived self-efficacy toward dengue prevention does not necessarily translate to the uptake of mosquito control measures. Understanding how these factors (knowledge and self-efficacy) influence mosquito control measures in Mexico is limited. Our study sought to bridge this knowledge gap by assessing individual-level variables that affect the use of mosquito control measures. A cross-sectional survey with 623 participants was administered online in Mexico from April to July 2021. Multiple linear regression and multiple logistic regression models were used to explore factors that predicted mosquito control scale and odds of taking measures to control mosquitoes in the previous year, respectively. Self-efficacy (β = 0.323, *p*-value = < 0.0001) and knowledge about dengue reduction scale (β = 0.316, *p*-value =< 0.0001) were the most important predictors of mosquito control scale. The linear regression model explained 24.9% of the mosquito control scale variance. Increasing age (OR = 1.064, *p*-value =< 0.0001) and self-efficacy (OR = 1.020, *p*-value = 0.0024) were both associated with an increase in the odds of taking measures against mosquitoes in the previous year. There is a potential to increase mosquito control awareness and practices through the increase in knowledge about mosquito reduction and self-efficacy in Mexico.

## 1. Introduction

Dengue Fever (DF) is caused by the most common arthropod-borne virus worldwide [1] and is transmitted through the bite of female *Aedes* spp. mosquitoes [2]. In Mexico, the transmission of DF occurs in 29 of 32 states [3], contributing to a cost of about $257 million each year [4,5]. DF results in longer waiting times in hospitals and increased financial burden, especially for people from low-income households [6]. Factors enhancing the distribution and spread of DF include vector dynamics such as the abundance of female *Aedes* spp., host preference, number of bites, environmental size, and the number of humans exposed to that environment [7]. Well-established preventive measures include using insecticide-treated curtains (ITCs) [8] and increasing individual knowledge about DF, how it spreads, and how to prevent it [9].

Effectively managing DF involves early detection and improved treatment, which helps reduce the dengue case fatality rate [10,11,12]. Interventions such as vaccination may prevent DF, but a more comprehensive public health intervention targets the source of the disease-causing organism. To achieve this, it is important to assess individuals’ willingness to engage in prevention efforts and determine factors that may affect their willingness and perceived ability to execute preventive measures. Prior studies demonstrated the relationship between knowledge, beliefs, and practices toward the prevention of DF [9,13,14]. While attitudes toward DF have been previously explored in areas such as Malaysia and Sri Lanka [13,14] self-efficacy, which directly influences health-seeking behaviors or adopting health behaviors, has not been explored in Mexico. Similarly, to the best of our knowledge, the association between these attitudes, self-efficacy, and a person’s history of practicing mosquito control measures has not been explored in Mexico. Moreover, while a person might be knowledgeable or self-efficacious, this might not directly affect his/her treatment and health-seeking behavior.

Despite well-known information on the spread and prevention of DF, there remains a knowledge gap about how dengue is transmitted. In parts of Sonora, Mexico, people infected with DF may be stigmatized because of beliefs that DF occurs due to person-to-person transmission [9]. Such beliefs may hamper DF prevention [9]. Furthermore, although people may know about DF generally, specific information such as DF symptoms, dengue hemorrhagic fever (DHF), and dengue shock syndrome (DSS) may remain unknown [15,16]. Additionally, awareness of DF transmission [9,17] and/or its relationship to factors such as climate change [18], does not guarantee adequate knowledge about domestic protection against mosquito bites nor positive attitudes toward DF prevention [15,19]. Multiple factors may shape an individual’s choices to engage in prevention practices, such as education level, age, and perceived susceptibility to DF [16,17].

In Mexico, DF research has mostly focused on warmer states in the southern regions [9,20]. However, up until 2014, cities in the north along the US Mexican border had not experienced local transmission [9]. Due to the limitation of studies in the north compared to the south of Mexico and the climatic diversity between the two regions, we hypothesized that knowledge and self-efficacy toward dengue prevention and its association with mosquito control would vary across regions in Mexico. Northern states such as Sonora on the US–Mexico border have recently reported an increasing number of dengue cases and deaths [9]. This study compared dengue awareness between geographically different regions (mainly Colima and Sonora) in Mexico. With the vast climatic differences across Mexico, behavioral attributes may vary as well.

## 2. Methods

### 2.1. Study Design and Respondents

This study used a population-based cross-sectional survey, which was distributed online across Mexico, with the majority of the targeted respondents living in Sonora and Colima. The survey was distributed to the professional, social, and personal networks of academicians in Colima and Sonora. Administration of the survey (Appendix A) was made between April and July 2021 and included eight sections. The survey contained a ‘demographics’ section, while the remaining sections assessed Knowledge, Attitudes, Beliefs, and Practices (KAP) related to mosquito control. Knowledge of DF prevention, transmission, symptoms, management, and its association with climate change was assessed. Respondents were then asked about their attitude toward education and mitigation strategies to lower the risk of DF. In the practice section, respondents were asked about household elimination methods, frequency of cleaning practices, presence of items conducive for breeding mosquitoes around one’s household, health-seeking behavior, and self-efficacy toward dengue prevention practices. This study was reviewed by the North Texas Regional Institutional Review Board and determined to be exempt from category research.

### 2.2. Study Areas

The study was conducted in representative urban and rural areas in mainly Sonora and Colima State, Mexico (Figure 1). These areas are expected to vary in their climate vulnerability, adaptive capacity, geographical and ecological diversity, and poverty. The study areas have a diversified climate with a dry season from October to April and a rainy season, with high rainfall, high humidity, and high temperatures from May to September in Colima. In contrast, rainfall is scarce between June and September in Sonora.

### 2.3. Instruments

A questionnaire was developed based on seven constructs and translated into Spanish. The seven sections were i. Socio-Demographics, which included information about age, sex, duration of living in an area, the area of residence (classified either urban or rural), kind of residence (this included options such as single-family homes, duplex, etc.), the state of residence, and the number of people living in an individual’s home, ii. ‘Knowledge about dengue fever’, which included questions about transmission, symptoms, and ways to access healthcare, iii. The ‘knowledge about climate change’ focused on whether individuals perceived there was a knowledge gap about climate change and its association with dengue fever, iv. Attitudes about climate change and dengue fever, v. The ‘practices related to dengue fever’ section contains questions about interventions taken by individuals to reduce mosquito breeding, vi. The ‘treatment-seeking approach and health-seeking behavior’ section assessed an individual’s access to healthcare, vii. The ‘level of self-efficacy towards dengue prevention practices at the individual, household level, and community level’ section focuses on individuals’ perception of their ability to carry out mosquito control measures.

### 2.4. Data Collection and Preprocessing

All study participants were given online consent forms which assured them of the confidentiality of the survey. After the forms were completed, they were sent the survey link to be completed online. The collected data were translated from Spanish into English for analysis. Content and construct validation were performed during pilot testing before the data collection. Variables were renamed in Microsoft Excel and exported into SAS (version 9.4, SAS Institute Inc., Cary, NC, USA) for data cleaning. Using G*Power 3.1.9.4, we determined that when performing a two-tailed hypothesis test for our regression model with an alpha of 0.05 and 3 to 5 predictors, a sample size of at least 262 was needed to determine an effect size as small as 0.02 [21]. The final sample comprised 622 respondents. This study was performed according to the Strengthening the Reporting of Observational Studies in Epidemiology (STROBE) reporting guidelines

### 2.5. Scoring Metrics

Metric scores and scales were created for knowledge about how dengue fever is transmitted (1–3), how to reduce dengue transmission (0–8), symptoms associated with dengue fever (0–9), climatic factors that affect dengue transmission (0–6), measures taken by respondents in the previous year to control mosquitoes (1–14), presence of items in respondent’s yard (0–8), the frequency of using the measure to control adult mosquitoes (0–5), and level of self-efficacy toward dengue prevention practices (14–140).

### 2.6. Analysis

Means were calculated for the score metrics and a confidence limit was derived for each. A chi-squared test was run to determine whether there were significant differences between dengue and climatic-related matrixes for different demographic variables. The alpha level was set to 0.05. A Pearson Correlation Coefficient (PCC) was performed to describe the relationship between knowledge, self-efficacy, and adult mosquito control. A final model was chosen to explore predictors of mosquito control measures, after satisfying the independence assumption (Durbin-Watson statistic = 1.934), ensuring normality (Shapiro–Wilk statistic = 0.987, *p*-value = 0.1060) and removal of influential points based on the formula CooksD > 4/n [22]. A multivariate logistic regression analysis was then used to determine variables associated with the odds of taking measures against mosquitoes in the previous year. Gender, age, and self-efficacy were fitted into the multivariate logistic models. The effect of each variable was analyzed, and the resulting models were compared using the likelihood ratio test.

## 3. Results

More than half of the participants in this study were female (65.92%) and the average age of respondents was 35 years. Overall, the differences in age across states was statistically significant ((F _(2, 618_) = 3.72, *p* = 0.0249). However, pairwise comparisons showed that this difference was only significant for the contrast between Colima and other states. A large proportion of the sample lived in Sonora (64.47%), followed by Colima (21.22%) and other Mexican states comprised 14.31%. An individual’s history of taking measures to eliminate mosquitoes within the last year varied statistically by the duration of stay in Sonora and Other states (Table 1). In Sonora, an individual’s history of taking mosquito elimination measures varied by age ((F _(2, 397_) = 17.17, *p* < 0.0001). This variance was not seen in Colima and other states.

### 3.1. Knowledge about Factors Associated with DF

Knowledge scores for DF transmission (Appendix A), symptoms (Appendix A), and climatic factors associated with DF (Appendix A) were higher for males. Although knowledge of DF reduction was higher in males for the total population (Appendix A) and Sonora (Appendix A), in Colima, females had more knowledge (Appendix A). In both Sonora and Colima, males had higher knowledge scores as regards the effect of climatic factors on dengue transmission than females (Table 2). This difference was statistically significant in Sonora. Males also had higher scores for the presence of items in their yards (Table 2 and Appendix A). Females had higher scores for measures taken to control mosquitoes in the previous year (Table 2) and a higher frequency of using measures to control adult mosquitoes (Table 2). Contrary to Sonora and Colima, in other states, females had higher knowledge scores and scored higher in taking measures to prevent mosquitoes in the previous year, while males had higher scores presence of items in the yard and a higher frequency of using measures to control mosquitoes (Table 2). Compared to Sonora residents, people living in Colima had higher scores for knowledge about how to reduce dengue transmission and dengue symptoms (Appendix A). Furthermore, unlike Sonora, in Colima and other states, people living in urban areas had higher knowledge scores compared to those in rural areas (Table 2). The group comparison was statistically significant in other states. For all states, urban areas had higher scores for measures taken to control mosquitoes and lower scores for the presence of items in the respondent’s yard. However, while rural areas in Sonora and other states scored lower for the frequency of using measures to control adult mosquitoes, rural areas in Colima scored higher (Table 2).

Across states, there was a statistically significant difference between states for being diagnosed (Appendix A) or having a neighbor diagnosed with DF in the past year (Table 3). The difference in having healthcare providers was significant between residence areas (Table 3).

### 3.2. Knowledge, Attitude, and Practices Associated with Climate Change and DF

People’s perceptions of the occurrence of climate change varied significantly depending on their area of residence (Appendix A). Similarly, there were significant variances seen by gender and state for perceived need for education programs on mitigation strategies related to climate change (Appendix A). States varied in their opinion about the need for governmental action against climate change and their use of larval and adult mosquito control measures (Appendix A). Compared to those living in rural areas, people living in urban areas had higher scores for taking measures to control mosquitoes in the previous year (Appendix A) and using adult mosquito control measures (Appendix A). Urban dwellers had lower scores for the presence of items in their yard (Appendix A)

### 3.3. Self-Efficacy Scores Associated with the Mosquito Reduction Practices

Figure 2A–C shows left skewness, indicating that the highest proportion of participants had higher self-efficacy scores. Although the minimum self-efficacy score for males was lower (14 vs. 17), their mean score was 118.54, with a median score of 124 while females had a mean score of 117.70, and a median score of 121 (Appendix A). The minimum self-efficacy score for Colima was 70, while that of Sonora was 14. The mean and median scores were also higher for Colima (121 and 125, respectively), compared to Sonora (117 and 121, respectively). The mean self-efficacy score for the urban areas (118) was higher compared to rural areas (114) (Appendix A).

Figure 3 shows no correlation between the self-efficacy scale and the adult mosquito control scale. However, there was a weak correlation (0.22) between knowledge about the dengue reduction scale and the adult mosquito control scale.

When looking at general mosquito control measures as a dependent variable in a multiple linear regression model (Appendix A), for every unit increase in the self-efficacy scale, there is a 0.04 increase in the mosquito control scale, accounting for other variables. Furthermore, when controlling for other variables, for every unit increase in the yard items scale, the mosquito control measure scale decreased by 0.21 units. After controlling for other factors, a one-unit increase in the knowledge about mosquito reduction scale increased the mosquito control measure scale by 0.25 units. Age was not significantly associated with the mosquito control scale (Appendix A). Of the four predictors in the final multiple linear regression model (Appendix A), self-efficacy had the highest standardized estimate, indicating that it was the most important predictor of mosquito control measures, followed by knowledge, then the presence of items in an individual’s yard.

Table 4 and Appendix A show the final multiple logistic regression model of factors that are associated with the odds of taking measures against the mosquito. Gender was not significantly associated with the odds of taking measures against the mosquito. However, a yearly increase in age was associated with 6% higher odds of taking measures against the mosquito. Moreover, a one-unit increase in the self-efficacy scale was associated with 2% higher odds of taking measures against the mosquito.

## 4. Discussion

Self-efficacy and knowledge were significant predictors of mosquito control measures. Higher scores of these predictors were associated more with Colima compared to Sonora. However, in this study, higher knowledge scores in Colima did not translate into a lower proportion of people who experienced DF in the previous year. Further exploration of these findings may require more objective measures of mosquito control in future studies and surveillance of DF to ensure cases are reported accurately across Mexico. However, to address inadequacies of mosquito control measures, programs may target ways to improve self-efficacy and knowledge barriers.

Both knowledge and self-efficacy may positively influence behavioral outcomes [23,24]. However, their effectiveness may depend on factors such as time constraints, risk perceptions, and trust [25,26,27]. Other studies found self-efficacy and trust to be more beneficial than the perceived risk of disease in predicting behavioral changes [28]. In our study, knowledge of DF reduction practices was weakly correlated with adult mosquito control measures (Figure 3), and this association was found to be significant in the multiple linear regression model. Similarly, a person’s self-efficacy to perform mosquito control measures was positively linked with mosquito control measures. Of the four predictors in the final multiple linear regression model (Appendix A), self-efficacy had the highest standardized estimate, indicating that it was the most important predictor of mosquito control measures, followed by knowledge, then the presence of items in an individual’s yard.

While people living in Colima had higher knowledge scores, DF diagnosis was highest in Colima. This unexpected negative association may be due to underreporting of DF in Sonora compared to Colima. Due to the longer presence of DF in the southern regions [9], northern states such as Sonora may be lagging on awareness and measures to take when presenting with symptoms of DF. Underreporting of these symptoms may further result in underdiagnoses [29]. Future studies may benefit from active surveillance of DF in Northern Mexico. On the contrary, the lower proportion of DF cases seen in Sonora might further explain the lower DF knowledge scores observed in Sonora. This implies that there may be a need to increase DF awareness in Sonora due to the implication of the *Aedes* spp. in the spread of DF in new locations and its adaptation to new environments [2]. Furthermore, as the climate changes, fluctuating temperatures lead to warmer northern states and cases increase northward [30,31], educational efforts will need to be intensified.

Males in Sonora and Colima had higher knowledge scores compared to females. However, neither gender nor the interaction between gender and knowledge scores was significant predictors of mosquito control practices. While higher knowledge scores were estimated for males, having items in one’s yard was also higher among males. This may be explained by women having a higher burden of household labor [32]. Hence, increased knowledge among women might reflect more preventive measures taken against mosquitoes. Higher frequencies of mosquito control measures in urban regions (Appendix A) were a consistent finding with prior studies which have shown higher adoption of mosquito control practices in semi-urban regions compared to rural regions [33].

Like its effect on mosquito control measures, increased self-efficacy was associated with increased odds of using control measures within the past year. Similarly, older age was also associated with higher odds of taking measures. This can be explained by the perceived low risk of DF among younger adult populations [16]. Compared to Colima, the age disparity between those who took mosquito control measures and those who did not is higher in Sonora; individuals who took measures against mosquitoes in the previous year were on average aged 37 in both Sonora and Colima. However, the mean ages for those who did not take measures in Colima and Sonora were 37 and 26, respectively. A further focus on this age disparity in Sonora may be warranted.

This study has some limitations. Notably, the survey was administered to a convenience sample by professional, social, and personal networks of academicians in Colima and Sonora. As a characteristic of convenient sampling techniques, the sampling frame was restrictive, with small sample size, limiting the generalizability of results to other populations. Overall, there were 32 extreme outliers identified when running the multiple linear regression model. However, these were excluded to prevent any source of bias in the estimates. Furthermore, based on our sample size calculation, our sample was large enough to significantly detect small effects on the linear scale. Due to the cross-sectional nature of the study, temporality may not be established, and causality may not be inferred from the estimates.

Future research may focus on programs that improve the knowledge and self-efficacy of individuals living in both Sonora and Colima. While knowledge scores were higher in Colima, a higher proportion of people diagnosed with DF indicates that there is still a need for mosquito control programs. Such efforts can include using game-based training courses used in insecticide resistance management [34]. Courses may be modified using citizen science for community-specific impact, which has been shown to improve mosquito control programs among individuals perceived to be self-efficacious [35]. Furthermore, using mobile applications in disseminating information about mosquito control may appeal to younger adults who had lower mean scores for taking measures against mosquitoes in the previous year.

## 5. Conclusions

This study assessed the association between knowledge about DF, self-efficacy, and the use of mosquito control measures in contrasting states such as Sonora and Colima in Mexico. It was observed that individuals in Sonora had lower scores in knowledge, as well as mosquito control outcomes compared to those in Colima. However, DF cases were higher in Colima. Age, self-efficacy, and knowledge were significantly associated with taking mosquito control measures. DF prevention interventions may focus on improving general knowledge about mosquito control measures in the female population and among younger individuals in Sonora. Future studies may use improved surveillance systems across states to mitigate potential underreporting. There should also be a focus on programs that help to improve an individual’s self-efficacy.

## Figures and Tables

**Figure 1 behavsci-12-00094-f001:**
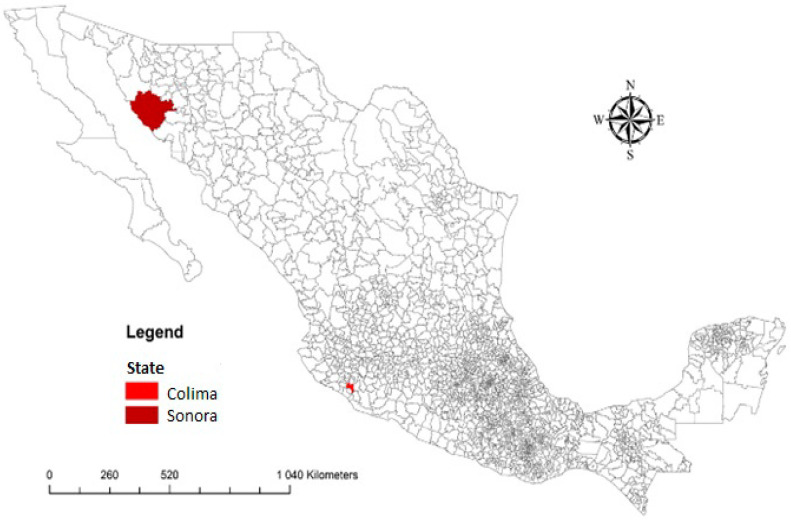
Map of Mexico showing locations of Colima and Sonora.

**Figure 2 behavsci-12-00094-f002:**
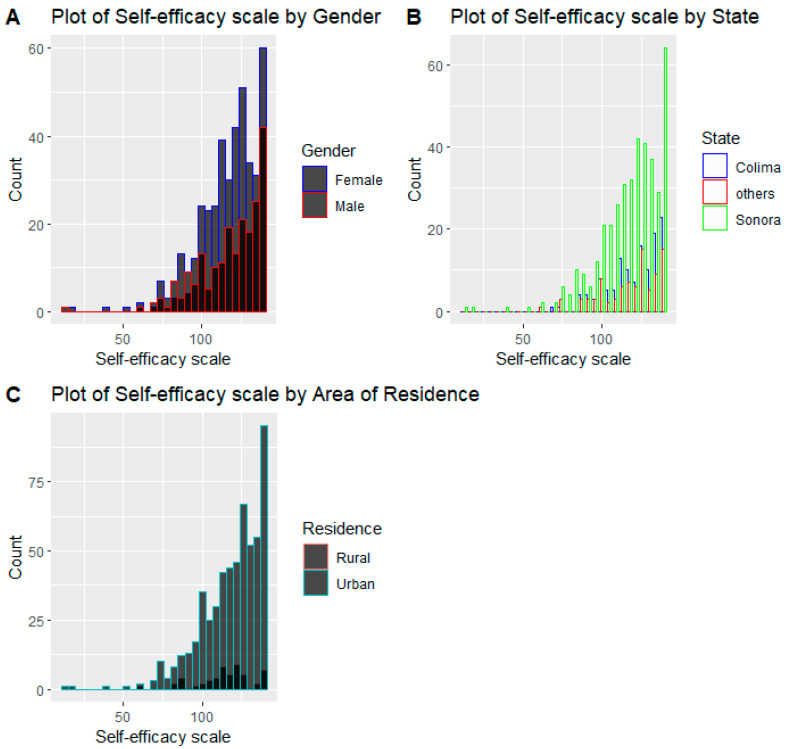
(**A**–**C**) Distribution of self-efficacy by demographic factors.

**Figure 3 behavsci-12-00094-f003:**
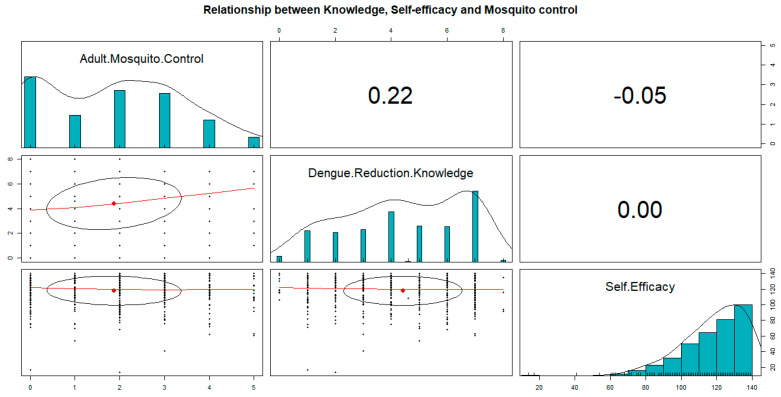
Pearson’s correlation of knowledge, self-efficacy, and mosquito control measure scales.

**Table 1 behavsci-12-00094-t001:** Demographic Characteristics by people who took measures to eliminate mosquitoes in their households in the past year.

States			Measures to Eliminate Mosquito (n (%))			
			Took Measures	Did Not Take Measures	Not Sure	*p*-Value
Colima	Age (years) ^a^		37.3 ± 11.7	37.4 ± 14.8	27 ± 4.0	
	Gender ^b^	Male	46 (86.79%)	6 (11.32%)	1 (1.89%)	0.2095
		Female	73 (92.41%)	3 (3.80%)	3 (3.80%)	
	Residence Area ^b^	Rural	6 (100%)	0 (%)	0 (%)	0.7094
		Urban	113 (89.68%)	9 (7.14%)	4 (3.17%)	
	Duration of stay ^b^	3 years or less	22 (88.00%)	1 (4.00%)	2 (8.00%)	0.2359
		Above 3 years	97 (90.65%)	8 (7.48%)	2 (1.87%)	
	Kind of Home ^b^	Single-family home	84 (89.36%)	7 (7.45%)	3 (3.19%)	0.8870
		*Others	35 (92.11%)	2 (5.26%)	1 (2.63%)	
	Number in Household ^b^	Less than 3	27 (87.10%)	3 (9.68%)	1 (3.23%)	0.8893
		3 or more	74 (90.24%)	6 (7.32%)	2 (2.44%)	
Sonora	Age (years) ^a^		37.1 ± 13.0	26.7 ± 8.6	29.7 ± 12.3	
	Gender ^b^	Male	104 (80.62%)	17 (13.18%)	8 (6.20%)	0.8633
		Female	225 (82.72%)	31 (11.40%)	16 (5.88%)	
	Residence Area ^b^	Rural	32 (78.05%)	6 (14.63%)	3 (7.32%)	0.7807
		Urban	297 (82.50%)	42 (11.67%)	21 (5.83%)	
	Duration of stay ^b^	3 years or less	32 (69.57%)	9 (19.57%)	5 (10.87%)	0.0620
		Above 3 years	297 (83.66%)	39 (10.99%)	19 (5.35%)	
	Kind of Home ^b^	Single-family home	254 (83.83%)	31 (10.23%)	18 (5.94%)	0.1639
		*Others	75 (76.53%)	17 (17.35%)	6 (6.12%)	
	Number in Household ^b^	Less than 3	61 (81.33%)	10 (13.33%)	4 (5.33%)	0.9851
		3 or more	170 (82.13%)	11 (5.31%)	26 (12.56%)	
ꭞOther States	Age (years) ^a^		33.2 ± 10.6	29.0 ± 11.6	24.0 ± 2.8	
	Gender ^b^	Male	26 (86.67%)	2 (6.67%)	2 (6.67%)	0.0509
		Female	48 (81.36%)	11 (18.64%)	-	
	Residence Area ^b^	Rural	5 (83.33%)	1 (16.67%)	0 (0.00%)	0.9216
		Urban	69 (83.13%)	12 (14.46%)	2 (2.41%)	
	Duration of stay ^b^	3 years or less	8 (57.14%)	4 (28.57%)	2 (14.29%)	0.0008
		Above 3 years	66 (88.00%)	9 (12.00%)	0 (0%)	
	Kind of Home ^b^	Single-family home	44 (84.62%)	8 (15.38%)	0 (0%)	0.2351
		*Others	30 (81.08%)	5 (13.51%)	2 (5.41%)	
	Number in Household ^b^	Less than 3	19 (82.61%)	3 (13.04%)	1 (4.35%)	0.9078
		3 or more	35 (85.37%)	5 (12.20%)	1 (2.44%)	

*Others includes structures such as Terrace, Duplex and Condominium. ꭞOther states include states such as Morelos, Nuevo Leon, Baja California, Chiapas, Ciudad de Mexico, Durango, Jalisco, Queretaro, Puebla, San Luis Potosi, Guerrero and Zacatecas. ^a^: Values were expressed as mean ± SD ^b^: Values were expressed as n (%).

**Table 2 behavsci-12-00094-t002:** Knowledge about factors associated with mosquito transmission, prevention, and control for different socio-economic parameters.

State	Variable		Knowledge about Climatic Factors Affecting Dengue Transmission	Measures Taken by Respondents in the Previous Year to Control Mosquitoes	Presence of Items in Respondent’s Yard	Frequency of Using Measures to Control Adult Mosquitoes
Colima	Gender	Male	2.34 (±1.29) (1.99, 2.69)	9.88 (±1.62) (9.21, 10.55)	1.85 (±1.29) (1.49, 2.21)	1.62 (±1.21) (1.29, 1.96)
		Female	2.06 (±1.54) (1.72, 2.40)	10.36 (±2.00) (9.58, 11.13)	1.79 (±1.42) (1.46, 2.13)	1.72 (±1.47) (1.39, 2.05)
		*T*-test *p*-value	0.2827	0.3482	0.8358	0.6853
	Residence Area	Rural	2.08 (±1.44) (1.16, 3.00)	10.00 (±2.06) (8.42, 11.58)	2.58 (±1.08) (1.89, 3.27)	1.75 (±1.86) (0.57, 2.93)
		Urban	2.18 (±1.45) (1.92, 2.45)	10.13 (±1.80) (9.60, 10.70)	1.73 (±1.37) (1.48, 1.99)	1.68 (±1.32) (1.44, 1.91)
		*T*-test *p*-value	0.8200	0.8148	0.0399	0.8570
	Kind of Home	Single-family home	2.20 (±1.42) (1.91, 2.49)	10.12 (±1.45) (9.6, 10.62)	1.82 (±1.43) (1.52, 2.11)	1.50 (±1.25) (1.24, 1.76)
		*Others	2.11 (±1.52) (1.61, 2.61)	10.16 (±2.41) (9.00, 11.32)	1.82 (±1.19) (1.41, 2.24)	2.13 (±1.55) (1.62, 2.64)
		*T*-test *p*-value	0.7285	0.9476	0.9701	0.0156
Sonora	Gender	Male	2.47 (±1.56) (2.20, 2.74)	9.41 (±2.33) (8.72, 10.11)	2.60 (±1.67) (2.29, 2.90)	1.78 (±1.52) (1.51, 2.04)
		Female	2.02 (±1.44) (1.85, 2.19)	9.66 (±2.32) (9.17, 10.16)	2.40 (±1.44) (2.22, 2.58)	1.90 (±1.55) (1.72, 2.09)
		*T*-test *p*-value	0.0049	0.5574	0.2538	0.4463
	Residence Area	Rural	2.23 (±1.43) (1.92, 2.55)	8.81 (±3.06) (7.60, 10.03)	2.53 (±1.39) (2.20, 2.84)	1.83 (±1.51) (1.50, 2.16)
		Urban	2.15 (±1.51) (1.98, 2.31)	9.77 (±2.06) (9.37, 10.17)	2.45 (±1.55) (2.27, 2.63)	1.87 (±1.55) (1.70, 2.03)
		*T*-test *p*-value	0.6381	0.0555	0.6975	0.8474
	Kind of Home	Single-family home	2.16 (±1.47) (2.00, 2.33)	9.59 (±2.22) (9.15, 10.03)	2.51 (±1.49) (2.33, 2.68)	1.88 (±1.50) (1.71, 2.05)
		*Others	2.16 (±1.55) (1.85, 2.47)	9.53 (±2.65) (8.58, 10.49)	2.35 (±1.61) (2.11, 2.68)	1.81 (±1.66) (1.47, 2.14)
		*T*-test *p*-value	0.9896	0.9013	0.3893	0.6889
Other States	Gender	Male	2.20 (±1.32) (1.71, 2.69)	9.22 (±2.39) (7.39, 11.05)	2.53 (±1.59) (1.94, 3.13)	2.27 (±1.39) (1.75, 2.78)
		Female	2.41 (±1.39) (2.04, 2.77)	9.87 (±2.00) (8.76, 10.97)	2.09 (±1.37) (1.72, 2.45)	2.20 (±1.57) (1.79, 2.61)
		*T*-test *p*-value	0.5023	0.4838	0.1788	0.8526
	Residence Area	Rural	1.82 (±1.37) (1.21, 2.42)	8.20 (±1.64) (6.16, 10.24)	2.43 (±1.83) (1.59, 3.26)	2.05 (±1.21) (1.51, 2.58)
		Urban	2.51 (±1.33) (2.18, 2.83)	10.00 (±2.11) (8.98, 11.02)	2.18 (±1.50) (1.92, 2.58)	2.18 (±1.55) (1.84, 2.51)
		*T*-test *p*-value	0.0391	0.0918	0.5073	0.5229
	Kind of Home	Single-family home	2.48 (±1.39) (2.09, 2.87)	9.80 (±2.11) (8.63, 10.97)	2.22 (±1.61) (1.76, 2.68)	2.29 (±1.53) (1.86, 2.71)
		*Others	2.13 (±1.32) (1.70, 2.57)	9.33 (±2.24) (7.61, 11.05)	2.28 (±1.23) (1.86, 2.70)	2.14 (±1.49) (1.64, 2.63)
		*T*-test *p*-value	0.2411	0.6131	0.8571	0.6386

**Table 3 behavsci-12-00094-t003:** Proportions and *p*-value for different Chi-Squared Test results with 95% significance.

	Variable	DF Diagnosis in the Past Year (n (%))			
Demographic Variable		Yes	No	Do Not Know	*p*-Value
Gender	Male	19 (8.96%)	192 (90.57%)	1 (0.47%)	0.5942
	Female	32 (7.80%)	373 (90.98%)	5 (91.22%)	
Residence Area	Rural	2 (3.77%)	51 (96.23%)	0 (0.00%)	0.3450
	Urban	49 (8.61%)	514 (90.33%)	6 (1.05%)	
State	Colima	29 (21.97%)	103 (78.03%)	0 (0.00%)	<0.0001
	Sonora	8 (2.00%)	388 (96.76%)	5 (1.25%)	
	Other states	14 (15.73%)	74 (83.15%)	1 (1.12%)	
	Variable	Availability of HealthCare providers (n (%))			
Demographic Variable		Yes	No	Do not know	*p*-value
Gender	Male	173 (81.60%)	17 (8.02%)	22 (10.38%)	0.2602
	Female	322 (78.54%)	27 (6.59%)	61 (14.88%)	
Residence Area	Rural	32 (60.38%)	10 (18.87%)	11 (20.75%)	0.0003
	Urban	463 (81.37%)	34 (5.98%)	72 (12.65%)	
State	Colima	101 (76.52%)	7 (5.30%)	24 (18.18%)	0.3421
	Sonora	322 (80.30%)	29 (7.23%)	50 (12.47%)	
	Other states	72 (76.52%)	8 (8.99%)	9 (10.11%)	
	Variable	Any neighbor diagnosed with DF (n (%))			
Demographic Variable		Yes	No	Do not know	*p*-value
Gender	Male	24 (11.32%)	86 (40.57%)	102 (48.11%)	0.7485
	Female	55 (13.41%)	165 (40.24%)	190 (46.34%)	
Residence Area	Rural	5 (9.43%)	28 (52.83%)	20 (37.74%)	0.1519
	Urban	74 (13.01%)	223 (39.19%)	272 (47.80%)	
State	Colima	31 (23.48%)	37 (28.03%)	64 (48.48%)	<0.0001
	Sonora	32 (7.98%)	177 (44.14%)	192 (47.88%)	
	Other states	16 (17.98%)	37 (41.57%)	36 (40.45%)	

**Table 4 behavsci-12-00094-t004:** The final model of factors associated with the odds of taking measures against mosquitoes.

Independent Variables	OR (95% CI)	*p*-Value
**Gender**		
Female	1.040 (0.603–1.792)	0.8890
Male	Ref	-
**Age**	1.064 (1.036–1.092)	<0.0001
**Self-efficacy**	1.020 (1.007–1.033)	0.0024

## Data Availability

Data will be made available based upon request through the corresponding author.

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
