# Peer review of "Determining Perceived Self-Efficacy for Preventing Dengue Fever in Two Climatically Diverse Mexican States: A Cross-Sectional Study"

_behavsci, 2022, doi:10.3390/bs12040094_

Round 1
Reviewer 1 Report
Please, see the attachment.

Reviewer 2 Report
In this study compared dengue awareness between geographically different regions (mainly Colima and Sonora) in Mexico. The study is interesting, but I have several comments. Furthermore, the manuscript is incomplete. Figures were not included.
Title: Consider the design "Determining Perceived Self-Efficacy for Preventing Dengue Fever in Two Climatically Diverse Mexican States: A Cross-sectional study"
Methods
Consider. This study was performed according to the Strengthening the Reporting of Observational Studies in Epidemiology (STROBE) reporting guidelines. I suggest including as an Annex, the check list STROBE.
Include a map of the study area.
Was there a sample calculation?
How was the questionnaire validated?
Was there a pilot test of the questionnaire?
Results
Estimate differences in age (years) with p-values.
Table 2. Why do not make intergroup comparisons?
Line 180. I suggest consider "DF diagnosis in past year, the availability of HealthCare providers, and any neighbor diagnosed with DF was statistically significant in Colima, compared to Sonora (2.00%) and other states".
The manuscript is incomplete. They have not included Figures 1, Figures 2a-d and Figure 3.
Table 4. Although there was an association, the OR for age and self-efficacy are very low. There's no probability.
The discussion section is well described.
Reviewer 3 Report
The paper would benefit from a clearer research question or argument around which it could be more clearly structured. The background section requires more discussion of the specific related research which would demonstrate the gap this research is filling. Several statements made in the paper are not supported by adequate empirical evidence or by making reference to relevant literature. Many statements appear in the discussion section without explanation as to the data on which they are based. The manuscript requires major revisions to contextualize the merits of the study and potential uses of its methodology in future studies. The main contributions of the paper should be presented as part of the empirical discussions or critical assessment on the core research outcomes. The recommendation regarding future research requires elaboration. Please provide more details regarding the study limitations and strengths and what this means for the study findings. The conclusions drawn are not well justified in the data collected and should clarify the main contribution of the paper and the value added to the field. A more discursive, analytical conclusion is needed, that engages with the theoretical questions in scholarship raised earlier in the paper.
The relationship between climate change mitigation and dengue fever in terms of knowledge and control measures has not been covered and thus such recent sources should be cited:
Ionescu, L. (2021). “Transitioning to a Low-Carbon Economy: Green Financial Behavior, Climate Change Mitigation, and Environmental Energy Sustainability,” Geopolitics, History, and International Relations 13(1): 86–96. doi: 10.22381/GHIR13120218.
Ionescu, L. (2021). “Leveraging Green Finance for Low-Carbon Energy, Sustainable Economic Development, and Climate Change Mitigation during the COVID-19 Pandemic,” Review of Contemporary Philosophy 20: 175–186. doi: 10.22381/RCP20202112.
Ionescu, L. (2020). “The Economics of the Carbon Tax: Environmental Performance, Sustainable Energy, and Green Financial Behavior,” Geopolitics, History, and International Relations 12(1): 101–107. doi:10.22381/GHIR121202010
Population's adherence to the dengue vaccine in comparison with COVID-19 vaccine hesitancy attitudes, behaviors, and perceptions has not been covered and thus such recent sources should be cited:
Popescu Ljungholm, D. (2021). “COVID-19 Threat Perceptions and Vaccine Hesitancy: Safety and Efficacy Concerns,” Analysis and Metaphysics 20: 50–61. doi: 10.22381/am2020213.
Morris, K. (2021). “COVID-19 Vaccine Hesitancy: Misperception, Distress, and Skepticism,” Review of Contemporary Philosophy 20: 105–116. doi: 10.22381/RCP2020216.
Lăzăroiu, G., Mihăilă, R., and Braniște, L. (2021). “The Language of Misinformation Literacy: COVID-19 Vaccine Hesitancy Attitudes, Behaviors, and Perceptions,” Linguistic and Philosophical Investigations 20: 85–94. doi: 10.22381/LPI2020217.
Reviewer 4 Report
All the information is described with little clarity. A new wording should be given to the content of the article, to sound more clearly and scientifically
Author Response
Thank you. The entire manuscript was reviewed and edited for increased readability. Please see the revised version of the manuscript with accepted and unaccepted changes.
Round 2
Reviewer 2 Report
No comments
Author Response
Thank you for your review
Reviewer 3 Report
The authors have failed to engage in important aspects of the various issues raised in the paper and have not provided sufficient (in some cases any) empirical or academic evidence supporting central and/or controversial points.
Reviewer 4 Report
Looks interesting to be publish!
Author Response
Thank you for your review and feedback!